

# Enriching the GEOFON seismic catalogue with automatic energy magnitude estimations

Dino Bindi[1], Riccardo Zaccarelli[1], Angelo Strollo[1], Domenico Di Giacomo[2], Andres Heinloo[1],
Peter Evans[1], Fabrice Cotton[1,3], and Frederik Tilmann[1,4]

[1]German Research Centre for Geoscience GFZ, Potsdam, Germany
[2]International Seismological Center ISC, Thatcham, UK
[3]Institute of Geociences, University of Potsdam, Germany
[4]Institute for Geological Sciences, Freie Universität Berlin, Germany

**Correspondence:** Dino Bindi (bindi@gfz-potsdam.de)

**Abstract.** We present a seismic catalogue including energy magnitude $M_e$ estimated from P-waves recorded at teleseismic distances in the range $20° \leq \Delta \leq 98°$ and for depths less than $80\,\text{km}$. The catalogue is built starting from the event catalogue disseminated by GEOFON,considering 6349 earthquakes with moment magnitude $M_w \geq 5$ occurring between 2011 and 2023. Magnitudes are computed using 1031396 freely available waveforms archived in EIDA and IRIS repositories, retrieved through standard FDSN webservices (https://www.fdsn.org/webservices/). A reduced, high quality catalogue for events with $M_w \geq 5.8$ and from which stations and events with only few recordings were removed forms the basis of a detailed analysis of the residuals of individual station measurements, which are decomposed into station and event specific terms, and a term accounting for remaining variability. The derived $M_e$ values are compared to $M_w$ computed by GEOFON and with the $M_e$ values calculated by IRIS. Software and tools developed for downloading and processing waveforms for bulk analysis and an add-on for SeisComP for real-time assessment of $M_e$ in a monitoring context are also provided alongside the catalogue. The SeisComP add-on is part of the GEOFON routine processing since December 2021 to compute and disseminate $M_e$ for major events via the existing services.

## 1 Introduction

Several magnitude scales have been defined to characterize the size of an earthquake. We can, however, divide magnitude scales in two groups: one including magnitudes based on the amplitudes and periods of different seismic phases measured on band-limited signals (e.g., the body- and surface-wave magnitudes,





Gutenberg, 1945b, a); the other including magnitude scales related to estimations of macroscopic physical parameters of the earthquake source. The latter comprise the moment ($M_w$, Kanamori, 1977; Hanks &
Kanamori, 1979) and the energy ($M_e$, Boatwright & Choy, 1986) magnitudes, which are based on seismic moment (Aki, 1966) and radiated seismic energy (Haskell, 1964), respectively. These two magnitude scales are somewhat complementary because, although both represent an estimation of earthquake-related energy, they are determined by different parts of the source spectrum. The seismic moment characterizes the low frequency end and represents the release of elastic energy stored in the Earth's crust or mantle,
being proportional to the integrated slip across the fault surface. The radiated seismic energy describes the fraction of the total released being radiated as seismic waves across all frequencies, i.e., it depends on the earthquake dynamics such as rupture velocity but also stress drop.

   $M_w$ is routinely computed from long period signals of broad-band recordings and it has become a robust and reliable source parameters for large and moderate earthquakes worldwide (Di Giacomo et al., 2021).
On the other hand the computation of $M_e$ is hindered by the necessity of integrating the velocity power spectra over a wide frequency range whilst using signals in a limited bandwidth and taking into account propagation effects at high frequencies.

   Aiming at validating and testing for operational purposes the procedures, we present a seismic catalogue of $M_e$ computed following the methodology proposed by Di Giacomo et al. (2008) and Di Giacomo et al.
(2010) for the rapid assessment of energy magnitude (i.e., without requiring additional source information other than the hypocentral location). The approach is based on the analysis of spectra computed for teleseismic vertical-component P-waveforms. We further present a detailed analysis of the residuals in a reduced high quality catalogue for events with $M_w \geq 5.8$ with respect to the $M_w$ available in the GEOFON catalogue and the $M_e$ values computed by IRIS.

## 2   Energy magnitude computation

### 2.1   Single station estimation

We implement the methodology proposed by Di Giacomo et al. (2008) and Di Giacomo et al. (2010) to compute $M_e$. Teleseismic vertical component P-waveforms (BHZ channels) are analyzed in the distance range from $20°$ to $98°$, and for earthquakes shallower than 80 km. Propagation effects are accounted for





by frequency-dependent amplitude decay functions, computed numerically (Wang, 1999) for the ak135Q model (Kennett et al., 1995; Montagner & Kennett, 1996) in the frequency range 0.012-1 Hz.

An estimate of radiated seismic energy $E_s$ is obtained for single station from the integral of the power spectra of the vertical component P waveform , corrected for propagation effects (Haskell, 1964):

$$E_s = \left[ \frac{2}{15\pi\rho\alpha^5} + \frac{1}{5\pi\rho\beta^5} \right] \int_{f_1}^{f_2} \left| \frac{\dot{u}(f)}{G(f)/2\pi f} \right|^2 df \tag{1}$$

where $\alpha$, $\beta$, and $\rho$ are the P-wave velocity, S-wave velocity and the density at the source, respectively; $f$ is the frequency and $f_1 = 0.012$ Hz and $f_2 = 1$ Hz are the lower and upper limits of the considered spectral bandwidth; $\dot{u}(f)$ is the P-wave velocity spectrum; $G(f)$ is the median value of Green's functions spectrum for displacement, which are computed across a wide range of plausible focal mechanism solutions and the median value is extracted.

We used analysis windows starting just before the P arrival and with lengths of 90 s for $M_w \leq 7.5$, 120 s for $7.5 < Mw \leq 8.5$ and 180 s for $M_w > 8.5$. The energy magnitude $M_e$ estimate for a single event station-pair is in turn computed as $M_e = 2/3(log_{10}E_s - 4.4)$, with $E_s$ given in Joule (Bormann et al., 2002). The procedure provides $M_e$ estimates at each recording station that can be averaged to minimize path-specific deviations not accounted for by the theoretical model (e.g., directivity and focal mechanism effects, regional variations in attenuation).

## 2.2 Open-source tool for computing $M_e$

The above procedure is implemented in the package *me-compute* (Zaccarelli, 2023). The program uses *stream2segment* (Zaccarelli et al., 2019; Zaccarelli, 2018) to download events, station metadata and waveforms from FDSN compliant repositories in a SQL database.

In our application, the download is configured to fetch events from the GEOFON (Quinteros et al., 2021) event web service, selecting events with computed $Mw$ in the time span 2011-2023. Waveforms are download from EIDA (Strollo et al., 2021) and IRIS ($https://service.iris.edu/$) data centers. The processing routine is implemented in a Python module which computes the station energy magnitude for each downloaded waveform segment, as summarized in section 2.1, and then calculates the event energy magnitude $M_e$ as the mean of all station magnitudes within the 5th–95th percentile range.

The final output consists of the following files:





- a tabular file in HDF format, where each row represents the metadata and measurements, specifically also the station energy magnitude estimate, for a single waveform.

- a tabular file in CSV format aggregating the results of the previous file, where each row represents a seismic event, reporting the event data end metadata, including the $M_e$ estimate for the event.

- an HTML file visualising selected content reported in the csv file, where the information for each event can be visualized on an interactive map

- one file per processed event in QuakeML format, where we included also the $M_e$ value.

All files produced by *me-compute* are disseminated in the data archive (Bindi et al. (2023); https://doi.org/10.5880/GFZ.2.6.2023.010), along with the *stream2segment* and *me-compute* configuration files.

## 3 Catalogue compilation

We use *me-compute* to compute $M_e$ for $M_w \geq 5$ earthquakes since 2011 in the GEOFON catalogue. Table 1 summarizes the steps followed to compile the disseminated $M_e$ catalogue. The catalogue reports the single waveform energy magnitude $M_{eij}$ estimated at station $j$ for earthquake $i$. The energy magnitude $M_e$ for each considered event $i$ is then computed as the median of $M_{eij}$ over the set of recording stations, without considering station static corrections. The starting data set D0 consists of more than one million waveforms (channels BHZ) generated by 6963 earthquakes recorded by 7765 stations belonging to 246 different networks. Only recordings with an average SNR for the amplitude greater than 3 within the frequency range of interest are included in D0. Several integrity and quality checks are applied to remove outliers and faulty signals. Data set D1 is obtained by analyzing the median residual at the network level, discarding 14 networks characterized by median residuals outside the 2.5 and 97.5 percentile range (Figure 1a). Data set D2 is then generated by analyzing the station median values and excluding 382 stations with residuals outside the 2.5-97.5 percentile range (Figure 1b). Most of the networks and stations removed will have instrumental problems or faulty metadata regarding instrument responses, although in some cases stations with very strong site effects might also be excluded.

The anomaly score, a classifier proposed by Zaccarelli et al. (2021); Zaccarelli (2022) is used to further refine the data set by flagging anomalous amplitudes. After inspecting the distribution of the anomaly scores, we set the threshold to 0.62 for $M_w < 7.5$ and to 0.80 for $M_w \geq 7.5$. The spatial distribution of

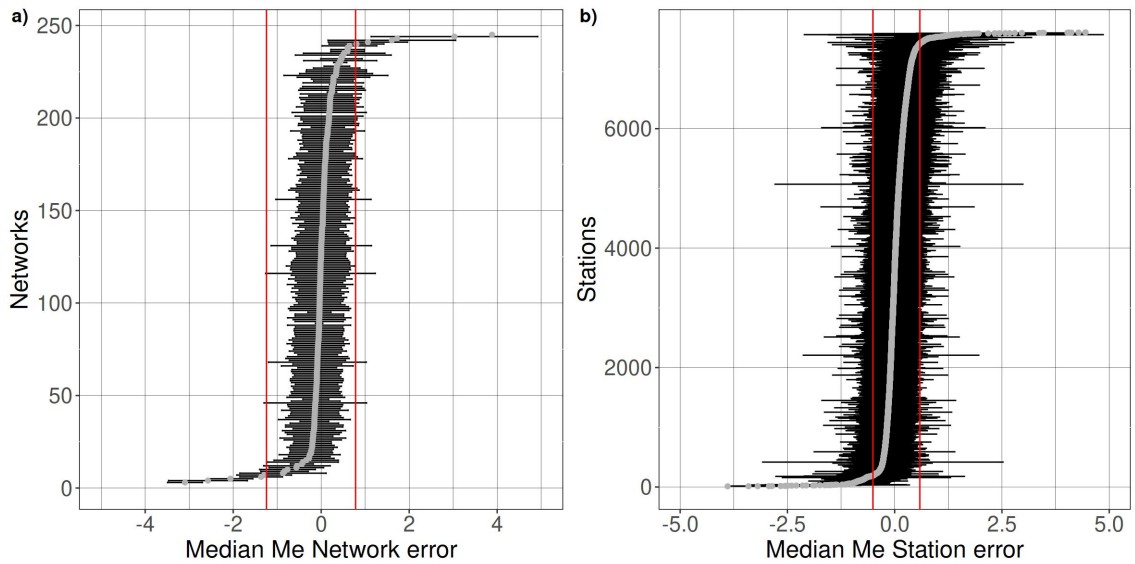

**Figure 1.** Median network residuals (circles) for data set D0 (left) and median station residuals for data set D1 (right); red lines correspond to the 2.5 and 97.5 percentiles of the distributions; for each network (left) and station (right) the horizontal bars correspond to the interval median (circle) $\pm$ 1 median absolute deviation (MAD). Few values falling outside the range considered for the horizontal axis are not shown.

events and stations generating the preferred (extended) data set D3 are shown in Figures 2a,b; this dataset
is disseminated as part of the supplementary dataset. The corresponding $M_e$ residuals are shown in Figure
3 against distance and $M_w$. The largest positive residuals correspond mostly to earthquakes with $M_w < 6$
recorded at distances $\Delta > 60°$, where the implemented methodology is expected to generate biased station
$M_e$ estimates due to the limitations in the analyzed bandwidth and low signal-to-noise ratio (Di Giacomo
et al., 2008, 2010). The overall residual distribution is unbiased and does not show trends of the mean
value with distance and magnitude.

Therefore, we further limit the dataset by only considering events with $M_w \geq 5.8$ and at least 10 single
station measurements; we further exclude stations with less than 10 recordings in total. We added a column
in the disseminated event dataset to flag lines fulfilling these further requirements; those flagged lines
correspond to data set D6, the final product of this study. It consists of $\sim 750000$ waveforms for 1671
earthquakes and 7135 stations. The event and station locations of D6 are shown in Figures 2c and 2d.

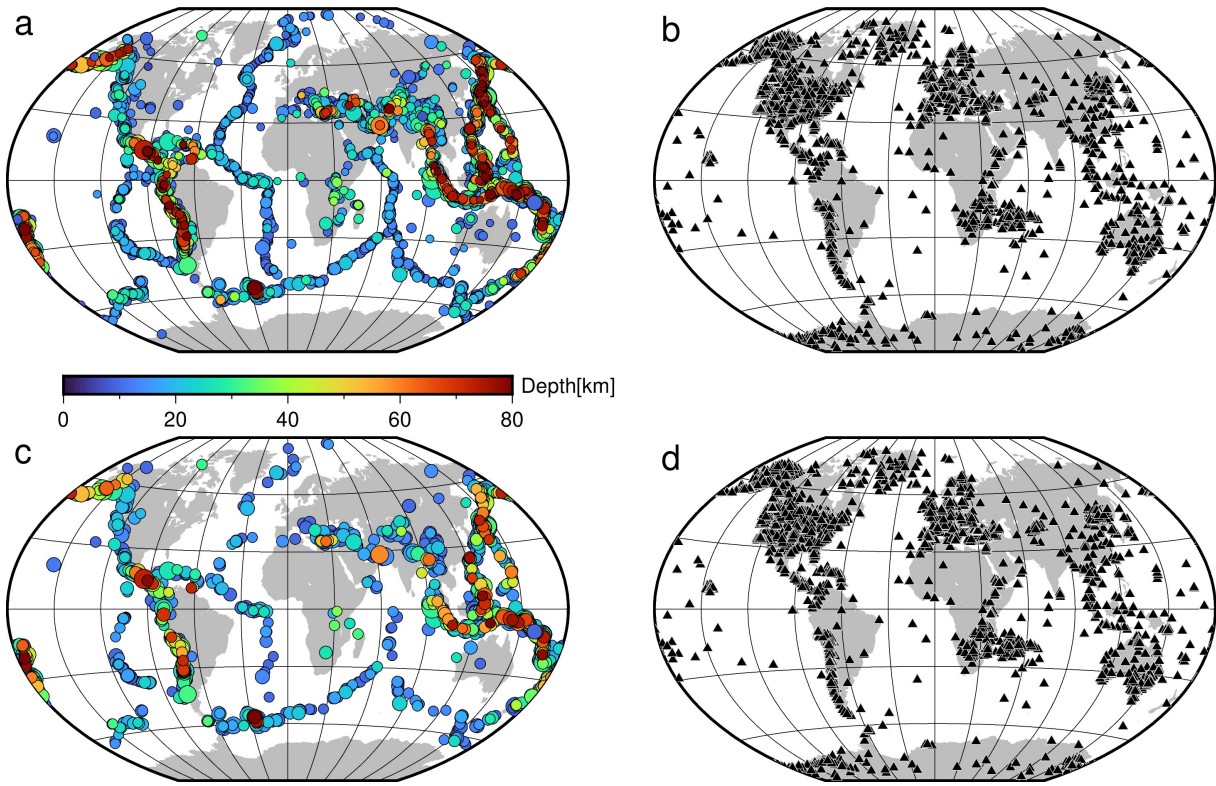

**Figure 2.** Panels a and b show event and station locations for data set D3 (Table 1), respectively; panels c and d show event and station locations for data set D6 (Table 1), respectively.

**Table 1.** Data sets considered in this study.

| Dataset | records | networks | stations | events | Selection |
|---------|---------|----------|----------|--------|-----------|
| D0 | 1126465 | 246 | 7765 | 6963 | $M_w \geq 5$ |
| D1 | 1072381 | 232 | 7617 | 6944 | Network selection (2.5-97.5 perc.) |
| D2 | 1034833 | 228 | 7235 | 6880 | Station selection (2.5-97.5 perc.) |
| **D3** | 1031396 | 228 | 7234 | 6349 | Anomaly score ($< 0.62, < 0.8$ for $M_w < 7.5, \geq 7.5$, resp. |
| D4 | 754025 | 228 | 7228 | 1731 | $M_w \geq 5.8$ |
| D5 | 751567 | 227 | 7135 | 1731 | #records per station $\geq 10$ |
| **D6** | 750903 | 227 | 7135 | 1671 | #record per event $\geq 10$ |
| Dg | | | | 153 | comparison between D6 and real time |

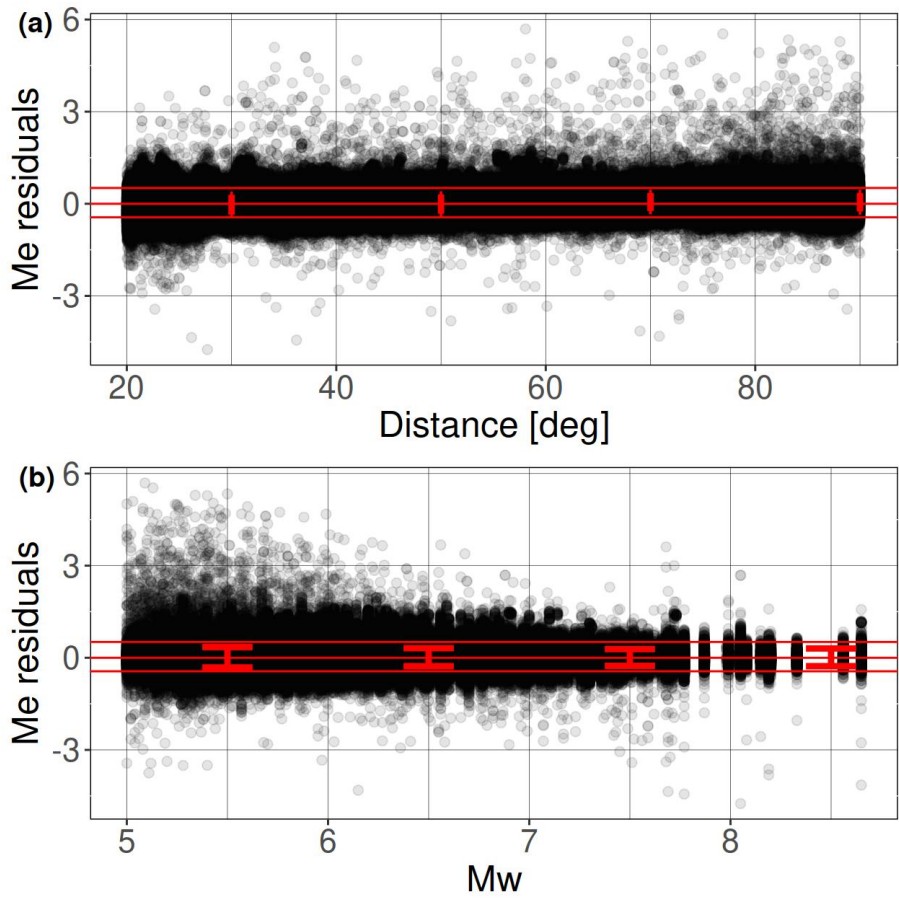

**Figure 3.** Energy magnitude residuals versus distance (a) and moment magnitude (b) for data set D3. The horizontal red lines bound the 90% confidence interval [-0.43,0.50] of the residual distribution; the error bars indicate the mean $\pm$ 1 standard deviation of the residuals computed over different distance (20° wide) and magnitude (1 m.u. wide) intervals.

## 4 Quality assessment via residual analysis

We perform residual analysis to validate the $D6$ catalogue. The relationship between $M_e$ and $M_w$ is analyzed by performing the following mixed-effects regression (Bates et al., 2015):

$$M_{eij} = c_1 + c_2 M_{wi} + \delta S_j + \delta E_i + \epsilon_{ij} \tag{2}$$

where $M_{eij}$ is the single waveform energy magnitude estimate at station $j$ for earthquake $i$; intercept $c_1$ and slope $c_2$ parameters define the median model; $\delta S_i$ and $\delta E_j$ are terms that capture station-specific and earthquake-specific adjustments, respectively; $\epsilon_{ij}$ accounts for the left-over effects (i.e., residuals that are

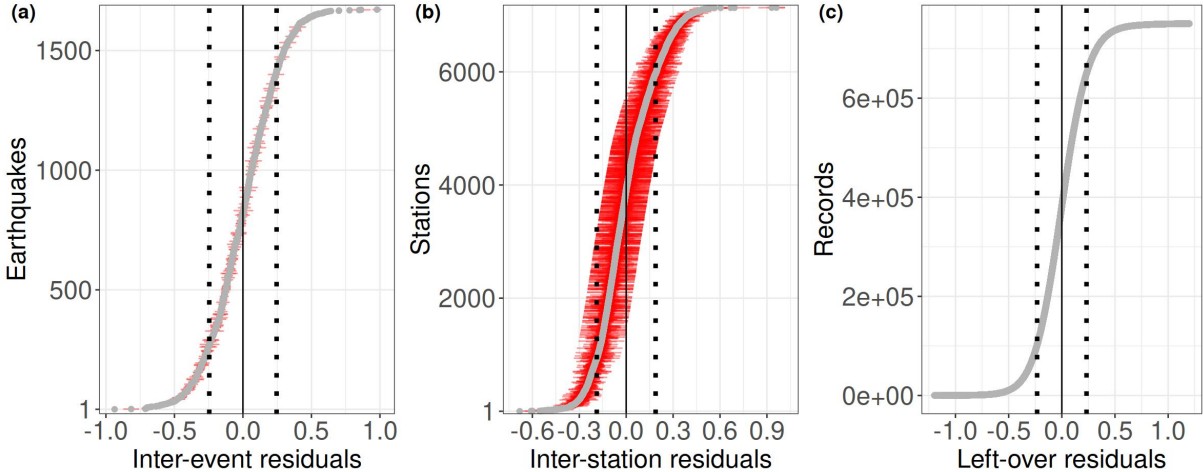

**Figure 4.** Cumulative distribution functions for event $\delta E$ (a), station $\delta S$ (b), and left-over $\epsilon$ distributions (circles) determined according to the mixed-effects regression in equation 2 applied to data set D6. Dotted lines correspond to standard deviations $\pm 1\tau$ (a), $\pm 1\phi_S$ (b, and $\pm 1\phi_0$ (c). Red horizontal lines in panels (a) and (b) are the standard errors of the random effects; in panel (c), values of $\epsilon$ exceeding $\pm 1.2$ in absolute value are not shown.

specific to a particular path/waveform). The random effects $\delta S$, $\delta E$ and $\epsilon$ are zero-mean normal distributions by construction. In particular, $\delta S_j$ (inter-station residual) can represent site effects or instrumental gain corrections, with most of the latter probably removed by the outlier filtering stages described above. The inter-event residual $\delta E_i$ is an event-specific deviation from the $M_e$ expected for a given $M_w$ from the linear regression term. Finally, $\epsilon_{ij}$ can be thought of as a noise term for individual measurements, which can be either related to path-specific heterogeneity in attenuation with respect to the 1D reference model, or the influence of ambient noise on the actual measurement.

The inter-event and inter-station term distributions are shown in Figure 4, which are described by standard deviations of $\tau$=0.27 and $\phi_S$= 0.19 m.u., respectively; the standard deviation of the $\epsilon$ is $\phi_0$=0.23 m.u. By combining the inter-event variability $\tau$ with the intra-event one equal to $\phi = \sqrt{\phi_0^2 + \phi_S}$, we obtain the total standard deviation $\sigma = \sqrt{\tau^2 + \phi^2} = 0.407$. Finally, the linear regression model is defined by coefficients $c_1$=(0.77 ± 0.09) m.u. and $c_2$=(0.92±0.01).

We show the spatial distribution of $\delta S$ in Figure 5. Since $M_{eij}$ is computed considering spectral values below 1 Hz, and using teleseismic recordings for distances above $20°$, $\delta S$ capture station-specific effects connected to large-scale geological and tectonic crustal features, as exemplified in Figure 5b 5 for stations located in Europe: positive $\delta S$ (i.e., $M_{eij}$ larger than the median) are observed for stations located in basins





**Figure 5.** (a) Distribution of the site-specific residuals $\delta S$, see equation 2 and (b) zoom over a portion of Europe. Numbers in (b) indicate the following locations: 1. Netherlands; 2 Harz highlands, Germany; 3 Switzerland; 4 Po plain, Italy; 5 Pyrenees mountain range; 6 Apennines mountain range; 7 East Anatolian fault region; 8 Moesian platform.





like in the Po plain, in the Moesian region, in the Netherlands, and in the East Anatolian fault region;
negative values $\delta S$ (i.e., $M_{eij}$ lower than the median) are observed for stations located in mountain ranges
such as the Pyrenees, the Alps, or in Harz highlands, but also tectonically highly active regions like the
East African rifts. The station terms can represent both site amplification, e.g. for stations in sedimentary
basins, and anomalously high or low attenuation in the crust and or mantle surrounding the station. The
station-specific residuals are disseminated along with the catalogue to allow the computation of $M_e$ for
future earthquakes taking into account static magnitude corrections to reduce variability.

The spatial distribution of the inter-event variability, $\delta E$, is shown in Figure 6 for the smallest and
largest values.

Considering depths shallower than 30 km (panels a and b), continental Asia, Philippines and Indonesia,
Aleutian islands show positive values; California, Mexico, central America, the Atlantic ridge are charac-
terized mostly by negative values. Considering deeper events (panels c and d), Japan and Philippines have
mostly positive values, Mexico and central America mostly negative values. The event specific residuals
are also disseminated along with the catalogue for increasing the usefulness of the product from the event
point of view and to allow the user to perform further refinements.

Path-specific residuals $\epsilon$ are shown in Figure 7 for three selected receiving areas in Europe, California
and Australia. Since in the partition of the residuals the left-over distribution $\epsilon$ represents the component
not related to systematic station and event effects, they are mostly connected to lateral variability in
attenuation in the Earth's interior with respect to the used global 1D model and amplitude variation related
to P wave radiation patterns for different focal mechanisms.

Finally, the $M_{eij}$ versus $M_w$ scaling defined by the linear regression coefficients $c_1$ and $c_2$ of equation 2
is shown in Figure 8.

### 4.1   Catalogue validation: comparison with IRIS

The energy magnitude computed in this study is compared to the values disseminated by IRIS through
the SPUD service IRIS DMC (2013). The methodology implemented by IRIS is described by Convers
& Newman (2011) and based on Boatwright & Choy (1986) and Newman & Okal (1998). Like us, the
energy flux is computed from the P-wave group (P+pP+sP) in the frequency domain. The single-station
estimations are corrected for frequency-dependent anelastic attenuation effects and converted back to the
energy radiated by the source by applying corrections for geometrical spreading, depth and mechanism-



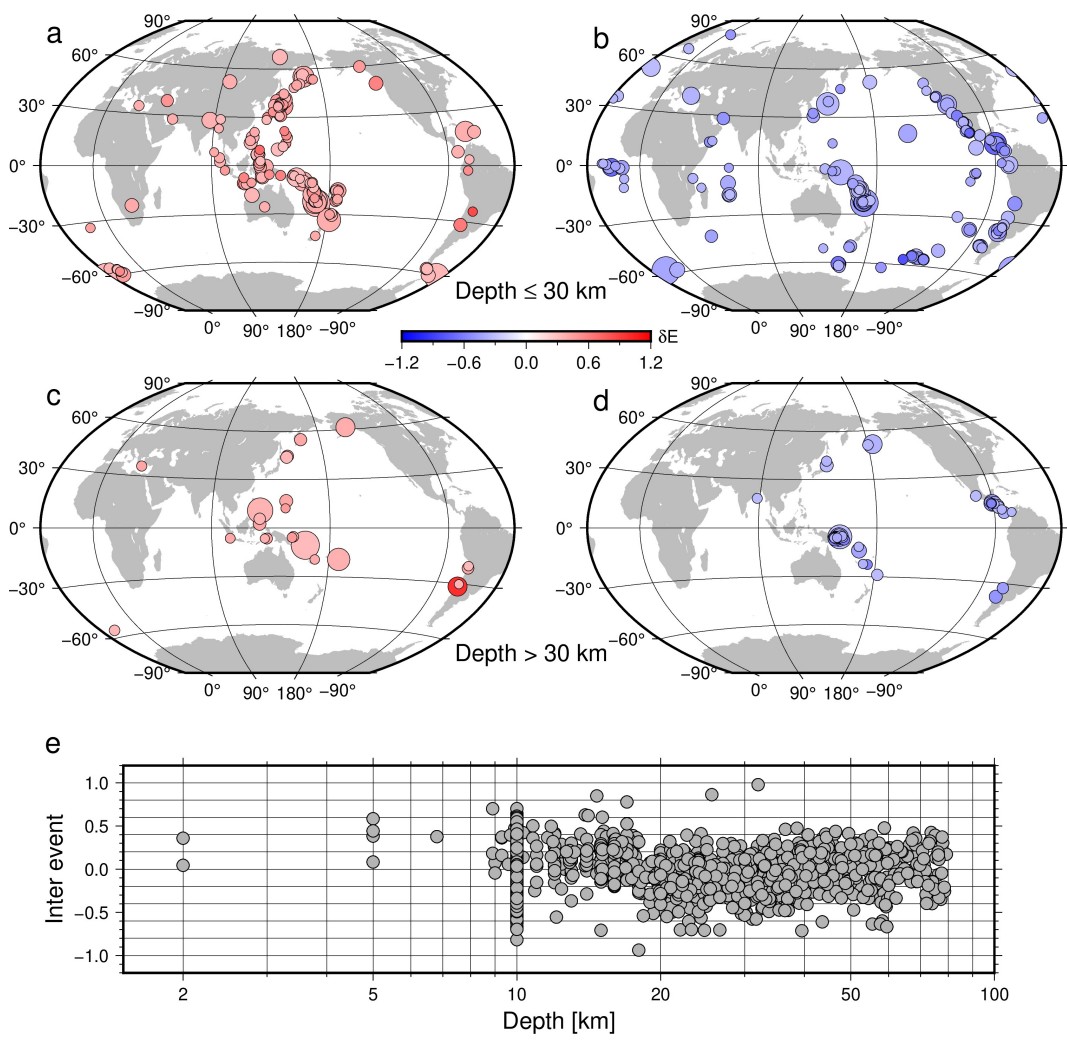

**Figure 6.** Extreme values for event specific residuals $\delta E$ for the $M_{eij}$ versus $M_w$ mixed-effects model of equation 2. Only values below the 10th percentile (panels b and d) and above the 90th percentile (panels a and c) of the distribution are shown (the percentiles are about $\pm\,0.3$). In panels a and b, earthquakes with hypocentral depths shallower than 30 km are selected; in panels c and d, events deeper than 30 km are considered. The distribution of $\delta E$ versus depth for all events is shown in panel e.

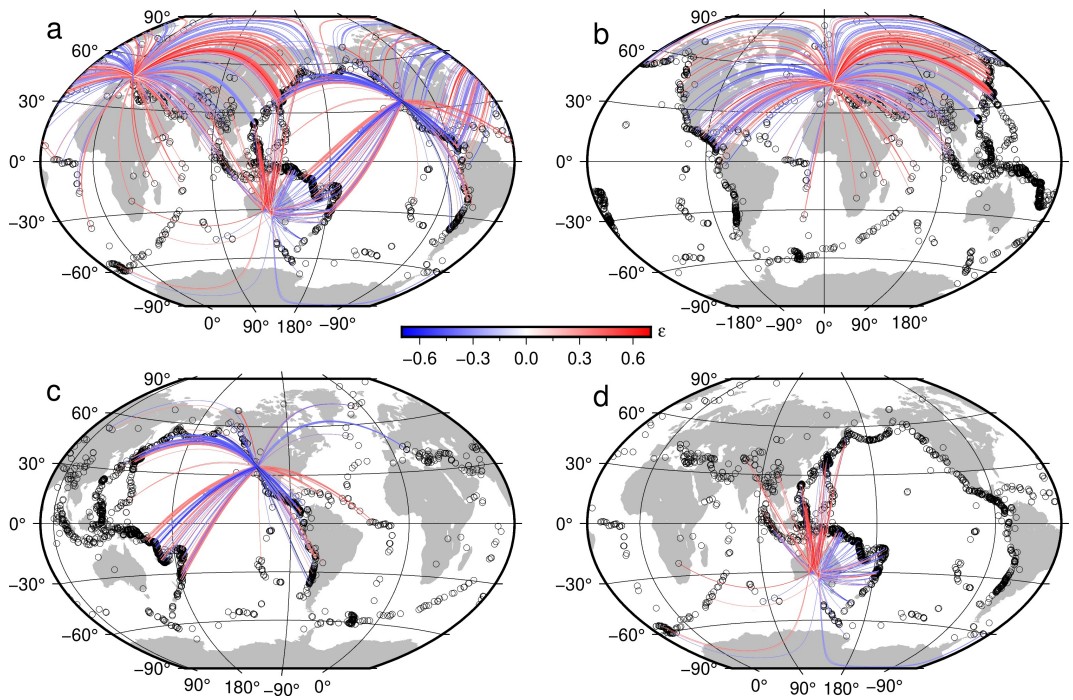

**Figure 7.** Left-over residual distribution $\epsilon$ of equation 2 2, showing only $|\epsilon| > 0.30$. a) residuals associated to three different receiving areas; b) as in panel a) but considering only the European receiving area; c) as in panel a) but considering only the receiving area in California; d) as in panel a) but considering only the receiving area in Australia. Circles indicate the earthquake locations.

dependent effects for P-waves, and considering a theoretical partition of the energy between P- and S-waves. The energy is computed considering the frequency range 0.014-2 Hz (broadband for $M_e(BB)$) or
0.5-2 Hz (high frequency for $M_e(HF)$), analyzing stations in the distance range $25° - 80°$. The duration of the time window used for the computation is based on analysis of the cumulative high-frequency energy (0.5-2 Hz) as a function of time. The crossover time used to compute the energy flux is identified at the intersection between the near constant increasing rate for short-times and the relative flat asymptotic behaviour for long duration. The SPUD service disseminates both the high-frequency $M_e(HF)$ and broad-
band $M_e(BB)$ estimates.

Two regression models are calibrated against the broad-band and high-frequency estimates disseminated by IRIS through SPUD: $M_e = -0.076 + 1.002M_e(HF) \pm 0.234$ and $M_e = 0.795 + 0.896M_e(BB) \pm 0.175$, as shown in Figure 9. For the magnitude range from 6 to 8, this results in biases of 0.06 m.u. for $M_e$ vs $M_e(HF)$, and varying from 0.17 to -0.04 m.u. for $M_e$ vs $M_e(HF)$, i.e., our estimates are nearly unbiased
relative to $M_e(HF)$ and tend to slightly overestimate $M_e(BB)$ at the lower end of the applicability range.

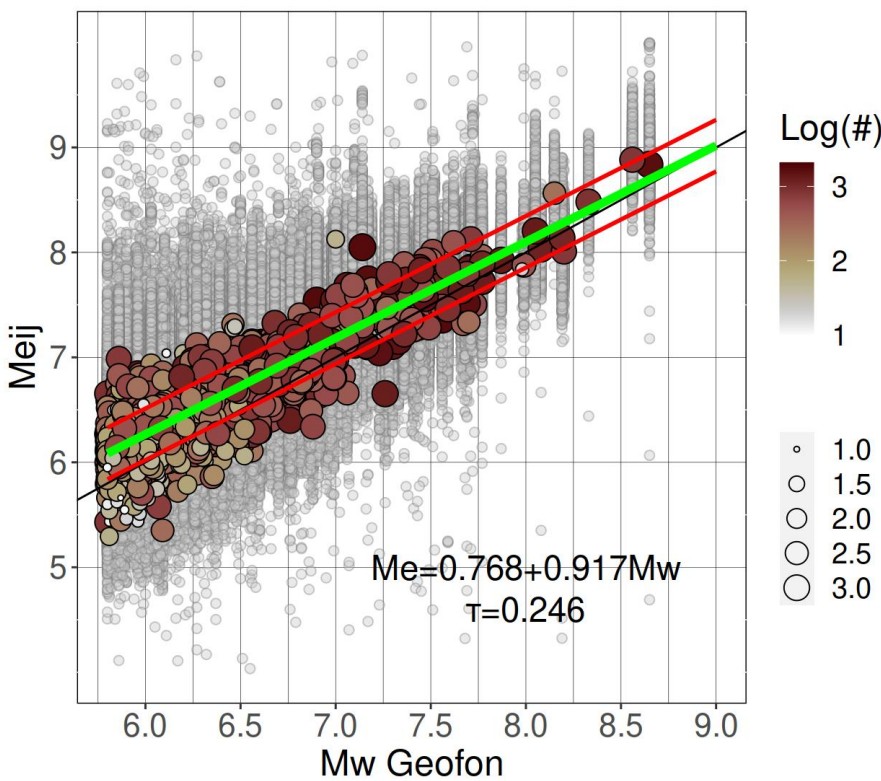

**Figure 8.** $M_{eij}$ versus $M_w$ scaling. Gray circles are the station $M_{eij}$ estimates, filled circles represent event $M_e$ values calculated as medians of all station estimates for that event; colour indicates how many stations contributed to each estimate. The best fit line in green is derived from the mixed-effects regression, equation 2, considering $\pm$ one inter event standard deviation $\tau$ (red lines). The faint black line shows equality for reference.

## 4.2 Catalogue validation: role of style of faulting

The faulting style is classified into normal, reverse and strike slip categories based on the plunge of the P,T and N axes (Frohlich & Apperson, 1992) as extracted from the GEOFON moment tensor solutions: normal fault(NF) if plunge(P) $\geq 60°$; strike slip (SS) if plunge(N) $\geq 60°$; thrust fault (TF) if plungeT $\geq 50°$. In the other cases, the earthquake is labeled with OF. To investigate the role of the style of faulting (SOF), we separate the event term into a fixed offset for each SOF class and a perturbation term for each event. If we indicate with $k = 1, 2, 3, 4$ the classes of the SOF grouping factor (corresponding to NF, SS, TF, and OF) and with $k_i$ the class of event $i$, the equation for the extended mixed-effects model is

$$M_{eij} = e_1 + e_2 M_{wi} + \delta S_j + [\delta SOF_{k_i} + \delta E_{SOF_i}] + \epsilon_{ij} \tag{3}$$

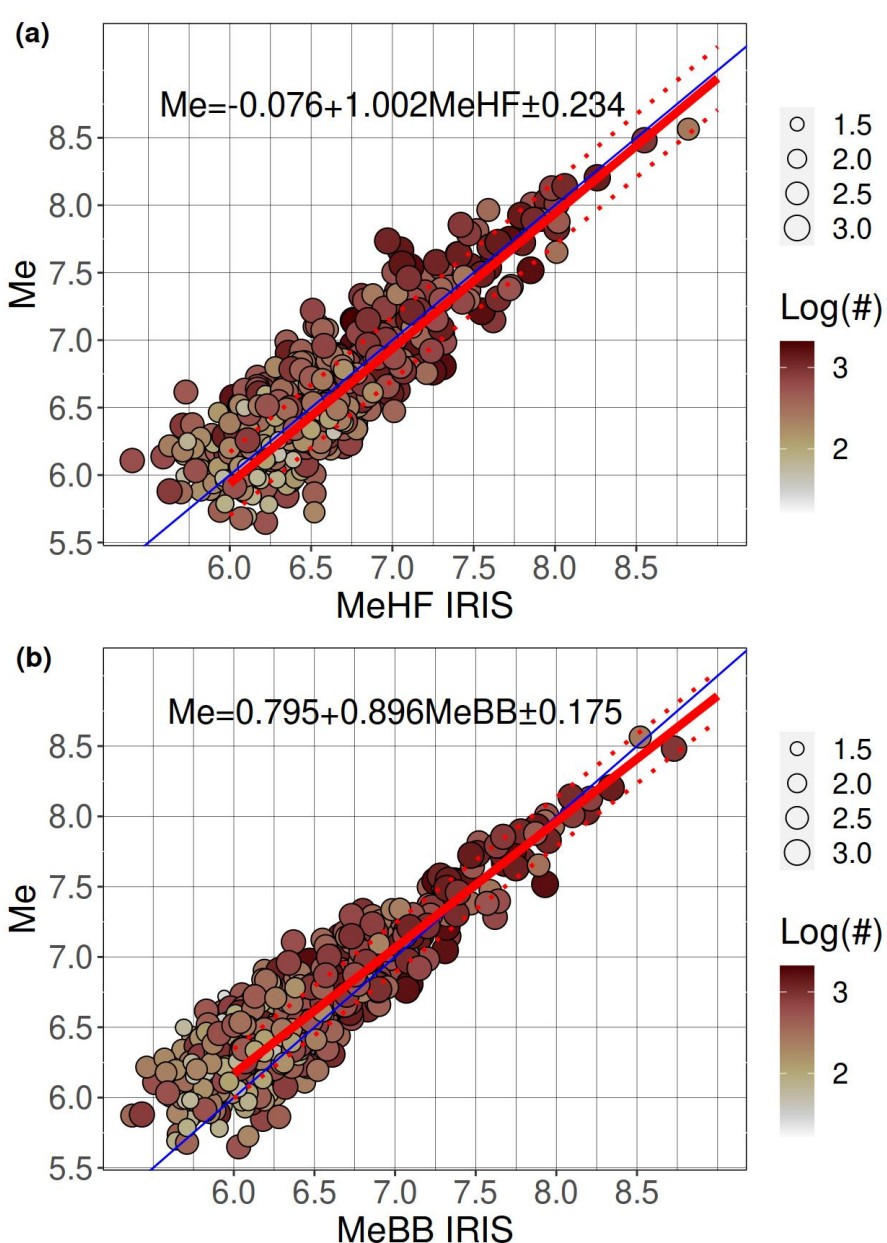

**Figure 9.** Comparison with energy magnitude disseminated by IRIS considering a) $M_e(HF)$ and b) $M_e(BB)$ (717 common events). The red line shows the linear regression fit, and the dotted lines show one standard deviation of the $M_e$ residuals. The blue line shows line of equality for reference.



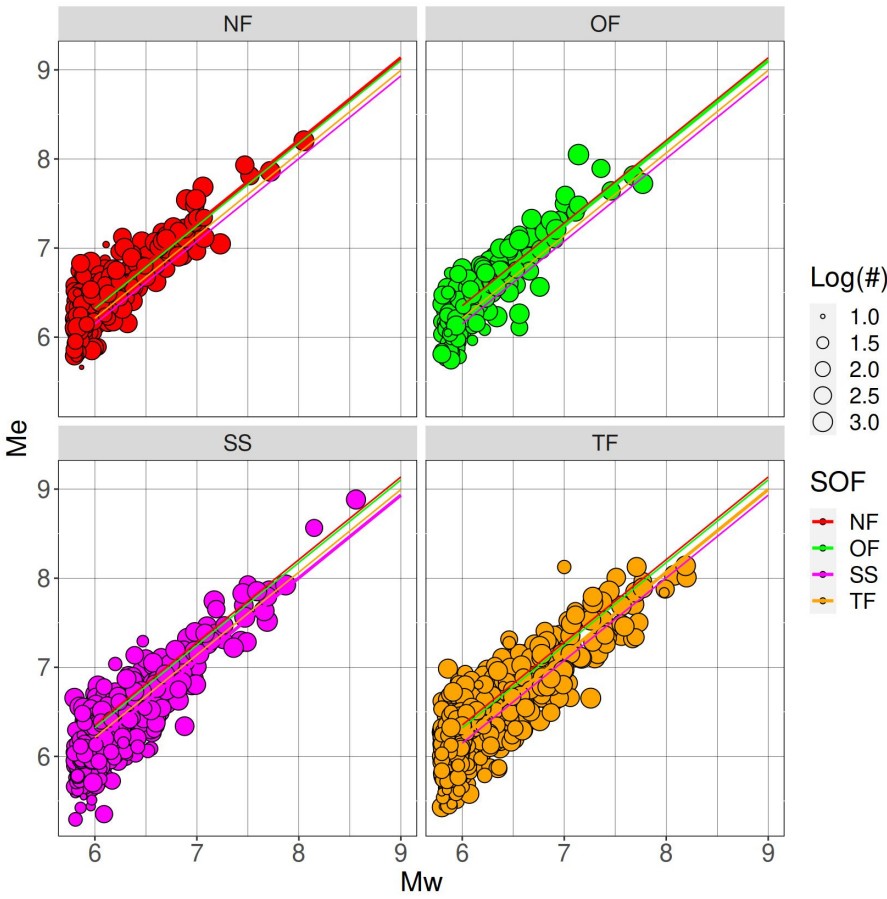

**Figure 10.** $M_e$ versus Mw categorized with SOF.

where $\delta SOF$ are the terms characterising the average effects of the the different SOFs and $\delta E_{SOF}$ are accounting for inter-event differences within each SOF class (nested random effects). The standard deviations of the $\delta S$, $\delta SOF$, $\delta E_{SOF}$ and $\epsilon$ distributions are $\phi_S = 0.190$, $\tau_{SOF} = 0.095$ $\tau = 0.236$, $\phi_0 = 0.232$, respectively, generating a total standard deviation $\sigma = 0.393$. The SOF terms are: $\delta SOF_1 = 0.098$ (NF), $\delta SOF_2 = -0.108$ (SS), $\delta SOF_3 = -0.045$ (TF), $\delta SOF_4 = 0.055$ (OF) (Figure 10). The largest difference is between SS and NF, in total 0.206 m.u.. There is a systematic impact of the SOF on the intercept of the model but associated variability is smaller compared to the inter-event variability $\tau$ (in other words, SOF effects are statistically significant but distributions of inter-event terms separated according to faulting style are strongly overlapping).





The SOF effects might arise due to physical differences (on average) between the different faulting
types, e.g., due to systematically different stress drops, differences in the maturity of faults or typical
environments (intra-plate vs interplate) that different faulting types occur most often, or they might be
artifacts due to the fact that the DiGiacomo method used here does not account for radiation pattern
effects, and the teleseismic arrivals utilised here sample preferentially certain parts of the focal sphere.
Therefore, we also investigate the role of the SOF in the relationship between $M_e$ derived in this study and
the $M_e(HF)$ and $M_e(BB)$ values disseminated by IRIS. We recall that the methodology implemented by
IRIS accounts for radiation pattern effects, which are related to the SOF. For this analysis, the regression
model is the following

$$M_e = g_1 + g_2 M_{iris} + \delta SOF + \epsilon \tag{4}$$

where $M_{iris}$ is either $M_e(HF)$ or $M_e(BB)$. Results shown in Figure 11 confirm that the largest intercept
difference is between normal and strike-slip events, and the differences in terms of m.u. are also similar
between the other SOF. This suggests that a large part of the SOF term is influenced by radiation pattern
effects, and interpretations of these differences in terms of geodynamics or hazard potential should be
done very cautiously.

## 5  Real-time module for SeisComP

The module, derived from *me-compute* has been integrated to the SeisComP package (Helmholtz Centre
Potsdam GFZ German Research Centre for Geosciences and GEMPA GmbH (2008)) and is part of the
GEOFON routine real-time processing since December 2021. The first event for which $M_e$ calculations
are available and disseminated via the usual GEOFON services is https://geofon.gfz-potsdam.de/eqinfo/
event.php?id=gfz2021xxzt, that occurred on 2021-12-07 10:28:00.3 UTC, ($M_e$ 5.7 and $M_w$ 5.5). The
*scmert* add-on is available at *https://github.com/SeisComP/scmert*.

The add-on has been configured at GEOFON to trigger the calculation for each origin created by the
automatic processing with magnitude $\geq$ 5.5, and to compute station magnitudes $M_{eij}$ for all stations/chan-
nels according to the definition of $M_e$ in the distance 20°-98°. The *scmert* procedure is applied with the
settings used by the GEOFON earthquake monitoring service, using stations available in real time from
the GEOFON Extended Virtual Network (https://geofon.gfz-potsdam.de/eqinfo/gevn/), including station-
selection and distribution trimming of 25%. The workflow for $M_e$ computations is as follows: as soon as





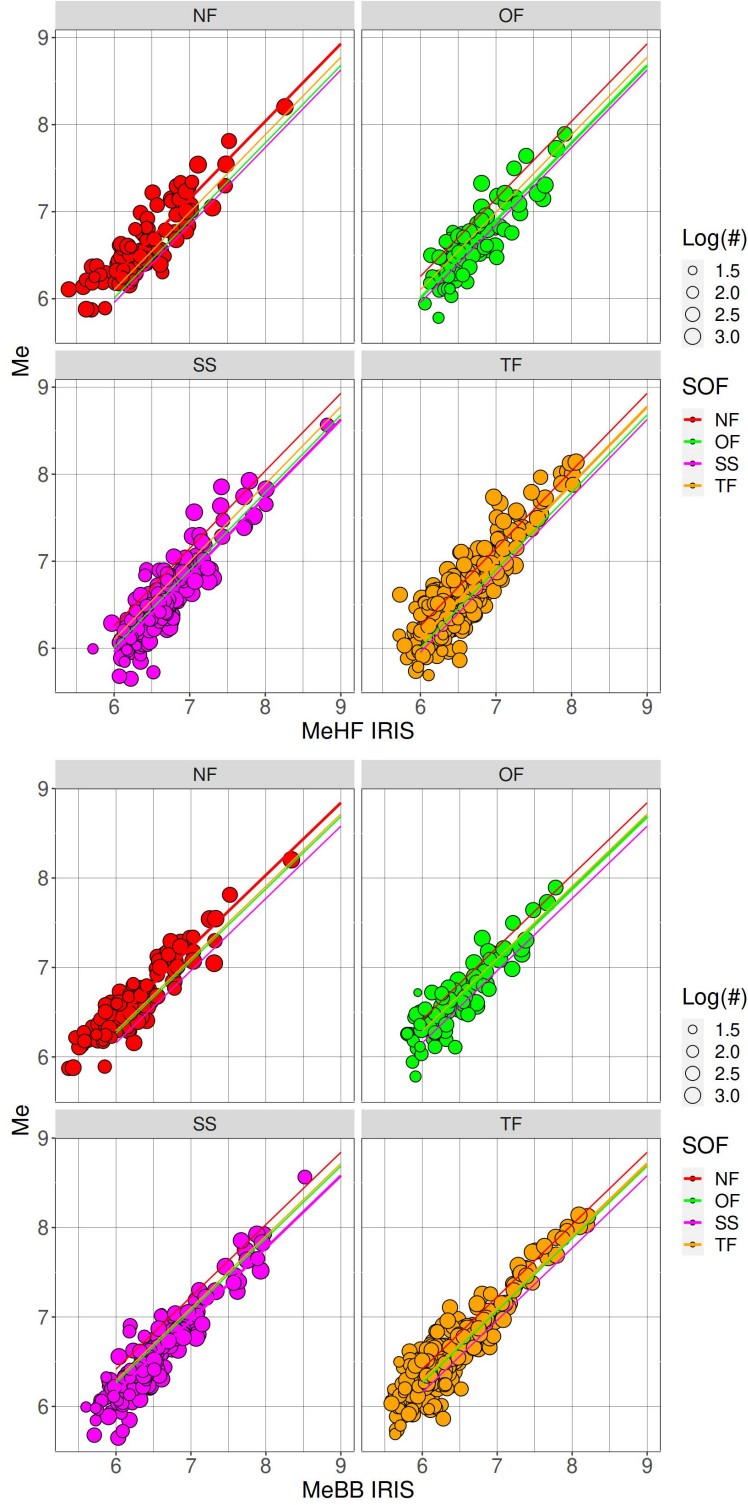

**Figure 11.** $M_e$ versus $M_e(BB)$ and $M_e(HF)$, categorized with SOF.





an automatically detected event reaches the magnitude threshold, *scmert* is triggered and starts to compute $M_{eij}$ upon receiving data from stations beyond 20°. The process continues until the selected window length (determined by the actual preliminary magnitude) of the last station at 98° is acquired. The first
estimate of the magnitude $M_e$ is released shortly after collecting 20 $M_{eij}$ estimates from individual station, usually within a few minutes of the earthquake's origin time. SeisComP modules continue to refine the estimate until no further updates are required (this includes manual release at later stages). The computed station magnitudes $M_{eij}$ are fully integrated also into the SeisComP Origin Locator View Graphical User Interface (*scolv GUI*, Figure 12) with station magnitudes and residuals displayed in a dedicated
energy-magnitude tab.

The energy magnitude values from both modules are compared in Figure 13. We used *scmert* with the same settings as the GEOFON earthquake monitoring service, including station selection and trimming of the distributions. The values are in good agreement, and the best fit model is $M_e = 0.057 + 0.987 M_e(GEO)$ with a standard deviation of 0.118. The average difference computed for magnitudes
between 6 and 8 is -0.028.

All values for $M_e$ that have been calculated since the start of the routine processing with *scmert* can be accessed via the fdsnws-event web service running at GEOFON by specifying Me as magnitude type (i.e., https://geofon.gfz-potsdam.de/fdsnws/event/1/query?starttime=2021-12-07&magnitudetype=Me& includeallmagnitudes=true&nodata=404). These values are also disseminated to other agencies (e.g. ISC, EMSC)
via the usual downstream channels, including real-time push service.

## 6 Code and data availability

Code used for computing the energy magnitude is available at:

  – off-line computations: *me-compute* https://doi.org/10.5880/GFZ.2.6.2023.008

  – real-time computations in SeiscomP: *scmert* https://github.com/SeisComP/scmert

Analyses have been performed in R (R Core Team (2020)) and we used the Generic Mapping Tools (Wessel et al. (2013)) to produce Figures 2, 5, 6, and 7. The archive including the energy magnitude catalogue (D3 and D6 in Table 1) and example of configuration files is available at: Bindi et al. (2023), https://doi.org/10.5880/GFZ.2.6.2023.010.





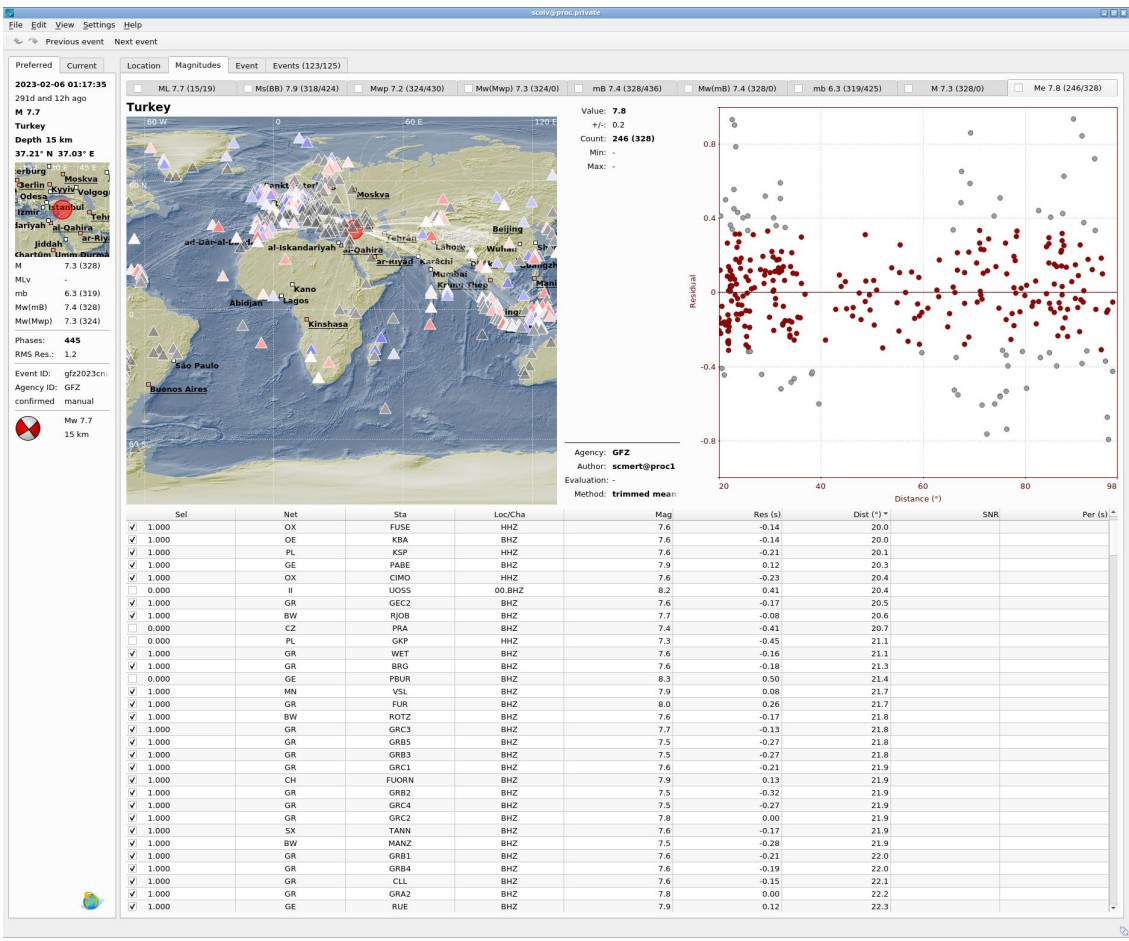

**Figure 12.** Screenshot of the SeisComP Origin Locator View (*scolv*) interactive tool to the Mw 7.7 Turkey earthquake, that occurred on February 6, 2023, 01:17 UTC along the East Anatolian fault. The obtained network magnitude value of $M_e$ is 7.8. Stations used are color coded according to Me magnitude residuals (top left frame), in gray stations excluded from the network magnitude not matching the distance range definition or trimmed while computing the average magnitude because within the +/- 12.5%. The top right scatter plot shows $M_e$ residuals by distance (in red those that contributed to actual $M_e$ network magnitude). The topography shown in the map is generated using the ETOPO1 global relief model (Amante & Eakins (2009)).





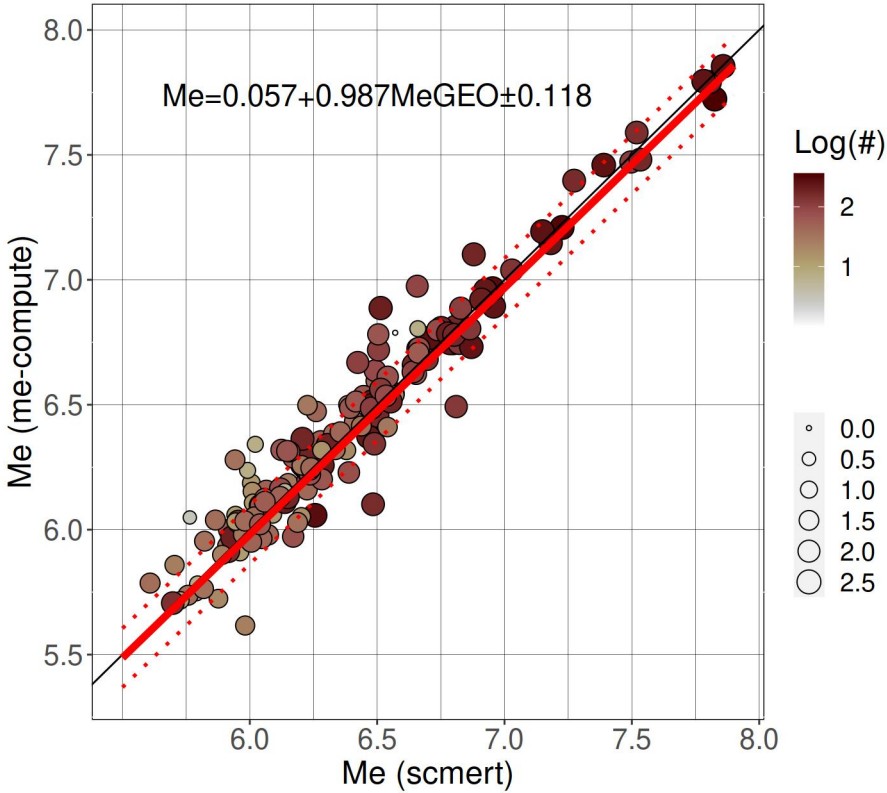

**Figure 13.** Comparison between $M_e$ computed in real-time by GEOFON with *scmert* add-on for SeiscomP (x-axis )and off-line estimation using *me-compute* (y-axis), considering 153 common events.

*Author contributions.* D.B., A.S. and D.DG. conceptualized the study; R.Z. developed the python code used to compile the disseminated

catalogue; A.H. developed the addon for SeiscomP; D.B. developed the quality checks; P.E., A. H. and A. S. organized the publication of $M_e$ by GEOFON via web-services; all authors participated to the finalization of the article.

*Competing interests.* The authors declare no competing interests.

*Acknowledgements.* We thank all network operators proving data via EIDA-ORFEUS and IRIS, as well as all real-time data providers contributing to the GEOFON virtual network. The complete list of references for the seismic networks analyze in this article with *me-*

255 *compute* is available at https://zenodo.org/records/10200493. The authors would like to acknowledge partial support from Horizon Europe Project Geo-INQUIRE, funded by the European Commission (HORIZON-INFRA-2021-SERV-01, project number 101058518).



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
