# Peer review of "Enriching the GEOFON seismic catalogue with automatic energy magnitude estimations"

_Earth System Science Data, 2023_

## Referee Comment (RC1)

Revision on: "Enriching the GEOFON seismic catalogue with automatic energy magnitude estimations" by Dino Bindi, Riccardo Zaccarelli, Angelo Strollo, Domenico Di Giacomo, Andres Heinloo, Peter Evans, Fabrice Cotton, and Frederik Tilmann

The paper aims to implement and extend back to 2011 the $M_e$ dataset furnished in real-time by GEOFON from December 2021. The importance of the energy magnitude in relation to the damage is known and having an extended database could be very useful for hazard studies.
In general, the goal is clear but some explanations about the used methodology are necessary to allow the publication.

Line 84: the final $M_{ei}$ for each event is computed as the median over the $M_{eij}$ of each station j. Why is it as computed as the median and not as some kind of average?

Line 96: The anomaly score is here introduced but some explanation of what it is, what the reported values mean, and how it is used to refine the dataset is needed.

Line 99: Why the preferred data set is also the extended one? What does extended mean in this case?

Line 100: From Fig. 3 a) and b) is hard to deduce that the residual analysis is unbiased and a trend is not present. The residual must be averaged over intervals of magnitude and distance (i.e., 0.1 m.u. and 1°) and plotted with the relative s.d. to show the lack of bias.

Line 114: The mixed-effect regression of eq. (2) is underdetermined because the number of unknown coefficients to be determined ($i+j+i \times j +2$) is larger tha the number of equations. I don't understand how it is possible to obtain all the parameters. The same underdetermination also holds for eq. (3).

Line 116: "*intercept $c_1$ and slope $c_2$ parameters define the median model*". What does it mean? $c_1$ and $c_2$ are not parameters obtained from the inversion of a matrix? How the parameter errors are calculated?

Line 127: $\phi = \sqrt{(\phi^2_0 + \phi^2_S)}$ ($\phi_S$ is square, check the text), and $\sigma=\sqrt{(\tau^2+\phi^2_0 + \phi^2_S)}$. I don't understand why to divide in two terms this calculation if it is the same as $\sigma=\sqrt{(\tau^2+\phi^2_0 + \phi^2_S)}=0.407$. Why the term 0.407 is not used anymore and in the relation reported in Fig. 8 the variability is only $\tau=0.246$ but in this case different from the previous one ($\tau=0.27$)?
Eq. 2 allows to calculate Me from Mw, what is the error on Me?

Line 174: "*varying from 0.17 to -0.04 m.u. for $M_e$ vs $M_e(HF)$*": $M_e(HF)$ is used in place of $M_e(BB)$. As both regressed variables are affected by errors of the same error a general orthogonal regression (GOR; Fuller, 2007; Castellaro et al., 2006), a squared error ratio ($\eta$) equal to 1 is more appropriate. What kind of regression was applied? What do the values 0.234 and 0.175 in the regression formulas correspond to? And also, what are the parameter errors?
The scaling of the obtained $M_e$ against SPUD $M_e(HF)$ seems to be close to 1:1. A simple statistical test (Student's t-test) could be useful to show if there is a significative difference from 1 of the slope for $M_e(HF)$ and also for $M_e(BB)$.

Line 200: Like the previous ones, the regressions of the equation (4) between $M_e$ for different faulting styles should be GOR (Fig. 11).

Line 233: Also in this case, a GOR is more appropriate.

Statistical analysis of the difference between the two types of $M_e$ could be useful to conclude that they are the same and the method proposed here could be implemented in real-time in the future providing an extended $M_e$ value dataset compared to the one currently on the GEOFON site.

The caption of Fig. 7: check equation 2 2.

Castellaro, S., F. Mulargia, and Y. Y. Kagan (2006). Regression problems for magnitudes, Geophys. J. Int. 165, 913–930, doi: 10.1111/j.1365- 246X.2006.02955.x.

Fuller, W. A. (1987). Measurement Error Models, John Wiley & Sons Inc., New York, 440 pp.

---

## Author Comment (AC1)

Revision on: **"Enriching the GEOFON seismic catalogue with automatic energy magnitude estimations"** by Dino Bindi, Riccardo Zaccarelli, Angelo Strollo, Domenico Di Giacomo, Andres Heinloo, Peter Evans, Fabrice Cotton, and Frederik Tilmann

The paper aims to implement and extend back to 2011 the Me dataset furnished in real-time by GEOFON from December 2021. The importance of the energy magnitude in relation to the damage is known and having an extended database could be very useful for hazard studies. In general, the goal is clear but some explanations about the used methodology are necessary to allow the publication.

We appreciate the Reviewer's valuable feedback and suggestions. Our responses to each comment are provided below.

Line 84: the final Mei for each event is computed as the median over the Meij of each station j. Why is it as computed as the median and not as some kind of average?

The median is more robust to outliers than the average. However, other options such as trimmed mean (as used in the SeiscomP application) are also possible. As the catalog reports also single station values, the end-user can apply other statistics to compute Me.

Line 96: The anomaly score is here introduced but some explanation of what it is, what the reported values mean, and how it is used to refine the dataset is needed.

An anomaly score is computed to further refine the data set by flagging anomalous amplitudes using the software sdaas (Zaccarelli et al. 2022). The software, developed from the work of Zaccarelli et al (2021) is based on a machine learning algorithm specifically designed for outlier detection (Isolation forest) which computes an anomaly score in [0, 1], representing the degree of belief of a waveform to be an outlier. The score can be used to assign robustness weights, or to define thresholds above which data can be discarded. We added this sentence to the manuscript around line 96.

Line 99: Why the preferred data set is also the extended one? What does extended mean in this case?

Following the reviewer's comment, we removed the term 'extended' as it could be misleading. It was used to indicate that D3 is the largest disseminated catalog to which further selections, such as limiting magnitude to values larger than 5.8, were applied. We decided to publish D3 as it is the largest data set after quality checks. We then applied further selections to D3 to produce D6 and flagged the entries corresponding to D6 in D3. As the end-user may wish to apply a different filter, we prefer to disseminate D3 while also specifying our selections for D6.

Line 100: From Fig. 3 a) and b) is hard to deduce that the residual analysis is

unbiased and a trend is not present. The residual must be averaged over intervals of magnitude and distance (i.e., 0.1 m.u. and 1°) and plotted with the relative s.d. to show the lack of bias.

As stated in the caption of Figure 3, the vertical error bars represent the residuals averaged over 1 m. u. and 20° (*Figure 3. Energy magnitude residuals versus distance (a) and moment magnitude (b) for data set D3. The 90% confidence interval [-0.43,0.50] of the residual distribution is bounded by the horizontal red lines, while the error bars indicate the mean ± 1 standard deviation of the residuals computed over different distance (20° wide) and magnitude (1 m. u. wide) intervals*). The reviewer has requested computing averages over denser intervals. However, increasing the density of the grid will not alter the message, as there are no discernible trends. This is demonstrated in the figure below, where the blue trend line was computed using a localized regression (loess method applied through the function geom_smooth of ggplot2, in R).

[Figure]

Line 114: The mixed-effect regression of eq. (2) is underdetermined because the number of unknown coefficients to be determined (i+j+i x j +2) is larger tha the number of equations. I don't understand how it is possible to obtain all the parameters. The same underdetermination also holds for eq. (3).

In mixed effects regressions, the parameters to be determined are the fixed effects (i.e. the model parameters) and the covariance matrix of the random effects, including the variance of the left-over residuals (in the case of equation 2, 5 quantities: $c_1$, $c_2$, tau, $phi_s$, $phi_0$). We performed the mixed-effects regression using the standard *lmer* function of R (Bates et al., 2015). For a detailed discussion of the mixed-effects regression and its application to the ground motion variability, please see Stafford (2014) [this reference has been added to the manuscript] and the

references therein.

Line 116: "intercept c1 and slope c2 parameters define the median model". What does it mean? C1 and c2 are not parameters obtained from the inversion of a matrix? How the parameter errors are calculated?

The parameters c1 (intercept) and c2 (slope) are the regression parameters determined through the mixed-effects regressions (the so-called fixed effects); when used for predictive purposes, they define the median model. Errors on the regression parameters are estimated from the asymptotic variances extracted from the covariance matrix of the fit.

Line 127: $\sigma = \sqrt{\phi^2_0 + \phi^2_S}$ ($\phi_S$ is square, check the text), and $\sigma = \sqrt{\tau^2 + \phi^2_0 + \phi^2_S}$. I don't understand why to divide in two terms this calculation if it is the same as $\sigma = \sqrt{\tau^2 + \phi^2_0 + \phi^2_S} = 0.407$. Why the term 0.407 is not used anymore and in the relation reported in Fig. 8 the variability is only $\tau = 0.246$ but in this case different from the previous one ($\tau = 0.27$)?

We thank the Reviewer for bringing the missing square to our attention. We corrected the value of tau in line 126.

Eq. 2 allows to calculate Me from Mw, what is the error on Me?

The standard deviation of 0.246 for the between-event residuals (random effects) can be used to quantify the uncertainty of Me from equation 2. It is important to note that due to the simplicity of the linear model and the large population of data used for the regression (~750000 data points), the uncertainty of the median model defined by c1 and c2 is very low. When evaluating the uncertainty of the median model using:

$$var\left[\bar{Me}\right]_{Mw} = J_o^T\left[varCov\right]J_o \text{ (eq\_a)}$$

which includes the Jacobian matrix ($J_o$) and the variance-covariance matrix (varCov), the standard deviation of the variance of Me regression in (eq\_a) for Mw=6 and 9 is 0.007 and 0.039, respectively.

Line 174: "varying from 0.17 to -0.04 m.u. for Me vs Me(HF)": Me(HF) is used in place of Me(BB).

Thanks, corrected.

As both regressed variables are affected by errors of the same error a general orthogonal regression (GOR; Fuller, 2007; Castellaro et al., 2006), a squared error ratio (━) equal to 1 is more appropriate. What kind of regression was applied?

A robust least squares regression was performed using the rlm function of R. In general, we agree with the Reviewer that total least squares regression (orthogonal

regression) could be advisable when accounting for uncertainties on both axes. Considering the high density of points in Figure 9 and assuming equal errors on both axes, the best-fit model obtained by performing an orthogonal regression (using the *odregress* function from the R library *pracma*, represented by the green line) is very similar to the ordinary least squares model (represented by the red line), as shown in the figure below for MeHF. We have chosen to maintain the results obtained with the robust regression.

[Figure]

What do the values 0.234 and 0.175 in the regression formulas correspond to?

They are the standard deviations of the residual distributions.

And also, what are the parameter errors?

We added the errors to our statement of the regression relations.

The scaling of the obtained Me against SPUD Me(HF) seems to be close to 1:1. A simple statistical test (Student's t-test) could be useful to show if there is a significative difference from 1 of the slope for Me(HF) and also for M e(BB).

For MeHF, a Student's t-test shows that the null-hypothesis that the slope is 1 cannot be rejected at 95% confidence (slope=1.0019, SE=0.0331, DF=363); for MeBB, the null hypothesis can be rejected (slope=0.8958, SE=0.0271, DF=363).

Line 200: Like the previous ones, the regressions of the equation (4) between Me for

different faulting styles should be GOR (Fig. 11).

Line 233: Also in this case, a GOR is more appropriate.

A mixed-effects regression using maximum likelihood is preferred, as it allows for the introduction of the SOF grouping factor to partition the residuals. Please, also refer to the previous answer for the MeHF regression.

Statistical analysis of the difference between the two types of Me could be useful to conclude that they are the same and the method proposed here could be implemented in real-time in the future providing an extended Me value dataset compared to the one currently on the GEOFON site.

The caption of Fig. 7: check equation 2 2.

Thanks, corrected.

---

## Author Comment (AC2)

In this article, the authors present a methodology to automatically compute Energy Magnitudes (Me). They apply the methodology to the GEOFON catalogue (2011-2023). The authors present several quality checks and statistical analyses of the dataset analyzed. They further compare their Me estimates with those made available by IRIS. The codes used both for off-line and real-time computations are made openly available.

The article if clear and well-written, and represents an important contribution by adding an additional magnitude estimation to the reference Geofon catalog.

We appreciate the Reviewer's valuable feedback and suggestions. Our responses to each comment are provided below.

I suggest only minor revisions below:

Abstract: Maybe add a brief sentence explaining the added value of Me estimates?

We added in the introduction (line 27) the sentence "Me estimates have been shown to play an important role when used in conjunction with Mw to better characterise the tsunami and shaking potential of an earthquake (Newman and Okal, 1998; Di Giacomo et al., 2010)."

Line 24: "… the low frequency end…", in practice we often measure Mw from the low frequency end of spectra, but really it represents the static (f = 0 Hz) component. Maybe just add that?

We substituted 'characterized' with 'extrapolated'.

L26: Correct to "… fraction of the total energy being radiated…"; "energy" is currently missing.

Thanks, we added 'energy'.

L 29: "parameter"; singular, not plural.

Thanks, corrected.

L 36: I'm not familiar with the methodologies to compute Me in detail, so I was a bit surprised to read that you compute Me from P waves, in opposition to S waves, which carry most of the energy. I guess it's related to the SNR. For the more unaware readers, maybe add a brief explanation on why you compute Me from P waves?

Papers illustrating methodologies to compute Me using teleseimic recordings go back to the 1980s (E.g., papers by Boatwrigth&Choy) and we feel that we do not need to repeat all the background but focus on our Me catalogue. However, we added the following after "vertical-component P-waveforms" at line 37 in the preprint: "(teleseismic P-waves are commonly used to compute Me for global earthquakes as their energy loss during propagation can be more reliably modeled compared to S-waves)"

L 44: Distance range: 20˚ to 98˚? I guess 98˚ is related to the P-wave shadow zone. Why disregard near source recordings? Add a brief explanation, again for the sake of the more unaware readers.

Similarly to the previous point we did not want to repeat the reasons for our setup because it largely follows what it is well established in the literature. However, we added the following after "98∘" at line 44 in the preprint: "(standard teleseismic range

usually starts at 30◦, but we use 20 to allow closer stations to be used for rapid response purposes. The shortest distances, however, are difficult to include for global earthquakes as regional effects are not be well accounted for with a global 1-D model)"

L 47: Suggestion: add "each" before "single station"
Added

Eq 1: Please double check this equation – is it dimensionally correct? Maybe I'm missing something…
We have double checked the equation, and also compared with other papers, such as Vassiliou and Kanamori, 1982, The energy release in earthquakes, BSSA. The equation appears to be correct.

L 53: Clarify what is "a wide range of plausible focal mechanisms"
We replaced ", which are computed across a wide range of plausible focal mechanism solutions and the median value is extracted" with "computed from multiple combinations of focal mechanisms, varying strike, dip and rake over regular grid (Di Giacomo et al. 2008)."

L 55: How much is "just before" the P wave arrival?
The configuration file of Me-compute allows for setting the number of seconds by which the starting time of the extracted window is shifted with respect to the theoretical P-wave arrival time. For our application, we used a 10-second shift [information added to the manuscript].

L 57: Correct to "a single event-station pair"
Done

L 66: Can you provide a rationale for starting in 2011?
The reason for this is that the Mw Geofon catalog starts from 2011.

L 86: Wouldn't you want to take into account static station corrections, once you've analyzed a large enough dataset? It seems like that would provide more robust Me estimates. It's a common correction when computing ML.
We acknowledge that station corrections can reduce the variance of magnitude computation. Therefore, we have provided station-specific residuals that can be used as station adjustments for future computations. The rationale behind the catalog compilation was to provide users with station magnitude values and all necessary information for computing and refining the event magnitude assessment.

L 87: 246 networks: Do you have a smart way to cite the DOIs of all those networks in your work??
The citations and DOIs provided as Supplement as written in the acknowledgments (https://zenodo.org/records/10200493) has been created writing a simple bash script running the IRIS service for citation (https://www.fdsn.org/networks/citation/). This information is provided in the supplement.

L 90: It was surprising to me that you find entire networks outside the 5-95 percentiles. Wouldn't it be enough to exclude stations outside the 5-95 percentiles? It's not very clear to me why you need to exclude entire networks.

The decision to remove certain networks was based on the fact that most of their stations were providing outlier values, likely due to incorrect or misused information in their station inventory files (e.g. units of generation constant).

L 96: "Anomaly score": maybe give a brief explanation on the grounds on which this method flags anomalous amplitudes?

An anomaly score is computed to further refine the data set by flagging anomalous amplitudes using the software sdaas (Zaccarelli et al. 2022). The software, developed from the work of Zaccarelli et al (2021) is based on a machine learning algorithm specifically designed for outlier detection (Isolation forest) which computes an anomaly score in [0, 1], representing the degree of belief of a waveform to be an outlier. The score can be used to assign robustness weights, or to define thresholds above which data can be discarded. We added this sentence to the manuscript around line 96.

Datasets D0, D1, …: it's not clear to me if you apply cumulatively or independently the quality criteria D1 -> D3. Please clarify.

The quality checks and selections indicated in Table 1 are applied sequentially in the order indicated [information added in the heading of Table 1, last column].

Figure 3: Can you overlay the dataset D6, in front of the black dots and behind the lines? In case the figure doesn't become illegible, it would be nice to see how much we lose from D3 to D6.

In terms of residual distribution with respect to Mw, only points above 5.8 were selected for D6, as shown in Table 1 (see the numbers). As for the distribution with respect to distance, it is difficult to distinguish due to the large number of overlapping points, with D3 having over 1 million points and D6 having about 750,000. It is worth noting that some of the large residuals are from Mw<6, as seen in the distribution with respect to magnitude, and therefore these values are not carried over to D6. We changed Figure 3.

[Figure]

L 127: Change to "with the intra-event, equal to…"
Agreed

L 127: phi = sqrt(…) , I believe a square (^2) is missing in the last parameter, phi_S.
Thanks, corrected.

L 137: East African Rift: active, but cratonic…
Thanks

L 149: "Similar to our approach" instead of "Like us"
Agreed

Fig 6e: Do you really need to use a log x scale? A big part of the plot is empty…
Done

[Figure]

L 166: "on the analysis"
Thanks

L 174: I believe it should be BB instead of HF, right before ", i.e.,"
Thanks

Figure 8: Very nice! It shows a lot of inter-station variability…
Thanks

L 180: Is it really 50 deg? All other styles of faulting have 60 deg.
Yes, see Frohlich & Apperson (1992)

L 180: Just write out that OF means "other faulting styles"
Done

L 196: ", where different faulting…"
Done

L 197: It was a surprise to read here that the method does not consider radiation pattern. When in line 52 you write that G(f) is computed for a range of plausible focal mechanisms, I though you took into account the focal mechanism, therefore the radiation pattern. Please clarify in the text. It seems like you should take into account the focal mechanism/radiation pattern, to get better Me estimates…
That's why we use median values from the Green's functions out of several computations from different focal mechanisms. The procedure is designed to be used without the knowledge of the focal mechanism, as, for example, already done by Newman&Okal, JGR 1998.

Fig 11: Maybe easier to read in a 2 rows x 4 columns plot? Top-BB, Bottom-HF.
We prefer to keep the grouping per magnitude type but we arranged the panels horizontally.

[Figure]

Finally, it would be really nice to have a "Conclusions" section, where you summarize the main take-away lessons from your new Me catalog. Why is this catalog useful? What new things are we learning from it?
We added a section 'Conclusive remarks' where we stated:
We computed the energy magnitude Me for 6349 events in the moment magnitude catalog disseminated by Geofon. When combined with Mw, Me allows for a better characterization of the tsunami and shaking potential of an earthquake. The procedure used to compile the data set, which includes 1031396 Me values for each recording station, is described in detail. Residuals are evaluated using a mixed-effects regression, which partitions the overall residuals into event-specific and station-specific contributions. These random effects are included in the distributed catalog, enabling the computation of Me for future events  using inter-station residuals as station corrections to reduce the uncertainty on Me. They also enable the assessment of energy magnitude adjustments for specific regions or faulting mechanisms by using inter-event residuals, and locating propagation anomalies with respect to the global model used to compute Green's functions using the left-over residuals. The methodology employed for computing Me (Di Giacomo et al, 2008) is suitable for the rapid assessment of Me (Di Giacomo et al, 2010). Therefore, it has been implemented as a module for SeiscomP, allowing for the automatic computation of Me in real-time and keeping the Me catalog up-to-date.

Great work! Thank you for this contribution.
Thank you and thanks for your comments.

---

## Author Comment (AC3)

Editor's comments.

In particular, following Review #1, I suggest you to:

- better clarify in the text which dataset corresponds to the provided DOI: the preferred is identified as D3 at line99, line 109 reads " D6, the final product of this study", and line 245 (in the data availability section) states that Bindi et al (2023) includes both datasets

The DOI is associated to D3; since D6 is a subset of D3, it is also included in the disseminated catalog. To fulfill the Editor's request, we removed "preferred" and 'final product". In the current version, these lines read as:

line 110: The spatial distribution of events and stations generating data set D3 are shown in Figures \ref{figure02}a,b; this dataset is disseminated as part of the supplementary dataset.

Line 118: We added a column in the disseminated D3 dataset to flag lines corresponding to D6

Line 278: The archive including the energy magnitude catalogue (D3 and D6 in Table 1) and example of configuration files is available at: \cite{Bindi23repo}, \ url{https://doi.org/10.5880/GFZ.2.6.2023.010}.

- briefly summarize the discussion on the uncertainty associated to Me and the selected regression method, including your reply to the reviewer's comments to Equation 2 and line 174.

Following the Editor's request, we have added a few lines on the variability of the single-station energy magnitude residuals and on the uncertainty of the mean values (lines from 135 to 145):

135 $\phi_0$=0.232 m.u.  Combining the inter-event variability $\tau$ with the intra-event variability equal to $\phi = \sqrt{\phi_0^2 + \phi_S^2}$, we obtain the total standard deviation $\sigma = \sqrt{\tau^2 + \phi^2} = 0.407$, which represents the variability of the single station $M_{eij}$ residuals with respect to the average $M_e$ computed per event. It is worth noting that the $\delta S_j$ values can be used as station corrections to compute the energy magnitude of new events. In this case, the inter-station contribution to the total variability is removed and the expected

140 variability of the $M_{eij}$ distribution is reduced to $\sqrt{\tau^2 + \phi_0^2} = 0.338$. Finally, the linear regression model is defined by the coefficients $c_1$=(0.77 ± 0.09) m.u. and $c_2$=(0.92±0.01). Considering the simplicity of the linear model in equation 2 and the large data set analyzed, the uncertainty on the median model (sometimes referred to as $\sigma_\mu$, Atik and Youngs, 2014 ) is very low, increasing from 0.007 for $M_w = 6$ to 0.039 for $M_w = 9$.

I also suggest using the same scale and grid spacing for both axes in the scatter plots (figures 8, 9, 10, 11 and 13).

At the suggestion of the Editor, we have tried to use the same scales for the figures indicated, but we prefer to keep the original versions. The motivation is that these figures show different quantities with values taken from different data sets. For example, below we report Figure 8 where we have extended the x-axis (where Mw is reported) from 5.8 to 5.25 to match the x-range in Figure 9 (where a different magnitude is reported, i.e. MeBB IRIS and MeHF IRIS). We do not like the graphical result, and considering that there is no particular scientific gain in aligning the ranges (because different quantities are shown), we prefer to stick with our original choice.

---

## Referee Report (RR1)

Final Revision on: "Enriching the GEOFON seismic catalogue with automatic energy magnitude estimations" by Dino Bindi, Riccardo Zaccarelli, Angelo Strollo, Domenico Di Giacomo, Andres Heinloo, Peter Evans, Fabrice Cotton, and Frederik Tilmann

The authors provided an explanation for each questions raised and added details and corrections to the manuscript that made some findings clearer, making it certainly suitable for publication. I would just like to ask the authors, if possible, to also include the two explanations (in blue) below in the manuscript before proceeding to publication. Thanks.

Eq. 2 allows to calculate Me from Mw, what is the error on Me?
The standard deviation of 0.246 for the between-event residuals (random effects) can be used to quantify the uncertainty of Me from equation 2. It is important to note that due to the simplicity of the linear model and the large population of data used for the regression (~750000 data points), the uncertainty of the median model defined by $c_1$ and $c_2$ is very low. When evaluating the uncertainty of the median model using:
$$\text{var} \, [\bar{M}e]_{Mw} = J_o^T [\text{varCov}] \, J_o \quad (\text{eq\_a})$$
which includes the Jacobian matrix ($J_o$) and the variance-covariance matrix (varCov), the standard deviation of the variance of Me regression in (eq_a) for Mw=6 and 9 is 0.007 and 0.039, respectively.

The scaling of the obtained Me against SPUD Me(HF) seems to be close to 1:1. A simple statistical test (Student's t-test) could be useful to show if there is a significative difference from 1 of the slope for Me(HF) and also for M e(BB).
For MeHF, a Student's t-test shows that the null-hypothesis that the slope is 1 cannot be rejected at 95% confidence (slope=1.0019, SE=0.0331, DF=363); for MeBB, the null hypothesis can be rejected (slope=0.8958, SE=0.0271, DF=363).